# Multimodal Low-light Image Enhancement with Depth Information

Zhen Wang*
Tokyo Institute of Technology
Tokyo, Japan
zhenwangrs@gmail.com

Dongyuan Li*
Tokyo Institute of Technology
Tokyo, Japan
lidy94805@gmail.com

Guang Li
Hokkaido University
Sapporo, Japan
guang@lmd.ist.hokudai.ac.jp

Ziqing Zhang
Shenzhen Ink Blue Technology
Shenzhen, China
zhangzq09@gmail.com

Renhe Jiang†
The University of Tokyo
Tokyo, Japan
jiangrh@csis.u-tokyo.ac.jp

## Abstract

Low-light image enhancement has been researched several years. However, current image restoration methods predominantly focus on recovering images from RGB images, overlooking the potential of incorporating more modalities. With the advancements in personal handheld devices, we can now easily capture images with depth information using devices such as mobile phones. The integration of depth information into image restoration is a research question worthy of exploration. Therefore, in this paper, we propose a multimodal low-light image enhancement task based on depth information and establish a dataset named **LED** (**L**ow-light Image **E**nhanced with **D**epth Map), consisting of 1,365 samples. Each sample in our dataset includes a low-light image, a normal-light image, and the corresponding depth map. Moreover, for the LED dataset, we design a corresponding multimodal method, which can processes the input images and depth map information simultaneously to generate the predicted normal-light images. Experimental results and detailed ablation studies proves the efficiency of our method which exceeds previous single-modal state-of-the arts methods from relevant field.

## CCS Concepts

• **Computing methodologies** → **Computer vision tasks**; **Image processing**; **3D imaging**.

## Keywords

Low-light image enhancement, Multimodal, RGBD, LiDAR camera

ACM Reference Format:
Zhen Wang, Dongyuan Li, Guang Li, Ziqing Zhang, and Renhe Jiang. 2024. Multimodal Low-light Image Enhancement with Depth Information. In

*Equal contribution.
†Corresponding author.

*Proceedings of the 32nd ACM International Conference on Multimedia (MM '24), October 28-November 1, 2024, Melbourne, VIC, Australia.* ACM, New York, NY, USA, 10 pages. https://doi.org/10.1145/3664647.3680741

## 1 Introduction

With the widespread use of smartphones, taking photos has become increasingly convenient and the number of images has grown exponentially. However, during photography, limitations imposed by device performance or environmental factors, including lighting conditions, time, and weather, can adversely affect the quality of captured photos. Restoring a low-quality image to a high-quality one has been a research focus in recent years. Image restoration encompasses tasks, such as removing artifacts [13], enhancing brightness [40], and eliminating effects like rain or snow [5]. In this paper, our primary focus is on the restoration of images captured under low-light conditions. The task of low-light image enhancement (LIE) [40] involves transforming dark photos taken in low-light environments into normal-light images. LIE has significant value in various scenarios, such as enhancing nighttime video frames and photographing in low-light conditions.

In recent years, with the advancement of deep learning technology, neural network models have been widely applied to LIE tasks. Currently the most widely used datasets include LOL (v1 [40] and v2 [44]), SID [4], SMID [3], SDSD [34], LIME [10] and DICM [21]. LOLv1 [40], in particular, is the earliest to propose a large-scale dataset specifically for the LIE task and introduces Retinex theory [19] for the first time into LIE. Subsequently, other researchers have proposed various methods for LIE. Some also leverage Retinex-based techniques [49, 50], which focus on separating the illumination and reflectance components of an image to enhance its visual quality. Others have explored the efficacy of encoder-decoder architectures [14, 43], coupled with attention mechanisms [2, 46], where attention-based mechanisms [33] play a pivotal role in directing the model's focus to relevant image regions.

Despite recent advances in the LIE task, the LIE task still has room for improvement, with a key consideration in the rapid developments in smartphones. Currently, modern smartphones are equipped with a greater number of sensors than a decade ago. Exploring how to leverage these sensors to enhance the restoration of photos taken in low-light conditions is a highly worthwhile research question. For instance, in current iPhone, LiDAR has become standard equipment, allowing for the easy capture of images with

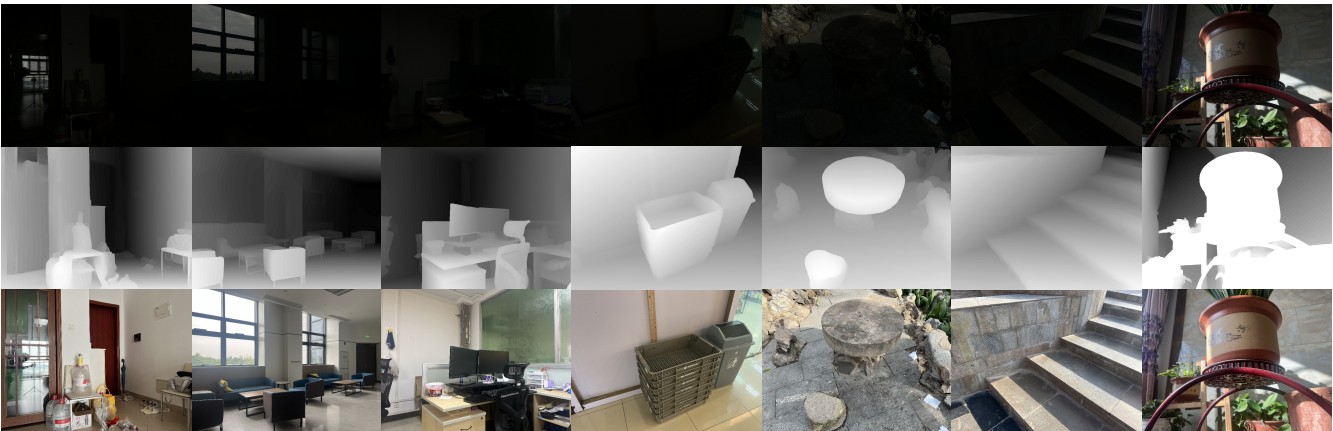

**Figure 1: Some samples from LED dataset. The first row is the low-light RGB image, the second row is the depth map (the darker colors indicate a greater distance from the camera), and the third row is the normal-light image.**

depth information by combining LiDAR with the camera. Therefore, it is important to research a method to integrate depth information into the LIE task for multimodal low-light image enhancement.

Given the absence of corresponding datasets, to facilitate research in this field, we first develop an application using the ARKit [1] provided by Apple. This application is able to capture depth information simultaneously while taking photos. We then use this application to create a dataset called **LED** (**L**ow-light Image **E**nhanced with **D**epth Map), which comprising 1,365 sets of images, including low-light images, normal-light images, and corresponding depth maps. Furthermore, in this paper, we propose a model called **LEDN** to effectively integrate depth information into the LIE pipeline. LEDN employs two methods to leverage depth information. The first method involves leveraging the clear and distinct edges in the depth map. Using the Sobel algorithm [16], boundary information is extracted from the depth map and then appended to the original RGB image. This could relive the issue of unclear and blur edges in low-light photos, and helps in segmenting the image into different self-correlated sub-regions. The second method involves the fusion of depth features, using an approach inspired by fast Fourier convolution [6] to merge features from RGB images and depth maps. Additionally, a module based on Mixture-of-Experts (MoE) is designed to fuse features from different decoder layers to more robustly and accurately generate the final prediction result.

With LED and LEDN, we conduct experiments and compare our method with previous state-of-the-art (SOTA) methods on the LIE task. The experimental results demonstrate that our method surpasses previous approaches in accuracy. Furthermore, we also carry out comprehensive ablation experiments, including both quantitative and qualitative analyzes, which confirm the effectiveness of our approach. Our contributions can be summarized as follows:

- We construct the first-of-its-kind large-scale low-light image enhancement dataset with depth information.
- We propose a novel method named LEDN, which could integrate depth information into LIE models, thus improving image restoration accuracy.

---

[1]https://developer.apple.com/augmented-reality/

- We demonstrate the effectiveness and flexibility of our method with detailed comparative experiments and ablation studies.

## 2 Related Work

### 2.1 LIE Datasets

Recent advances in low-light image enhancement have spurred significant interest in developing datasets that accurately reflect the challenges posed by dark environments. One notable and earliest dataset in this domain is the LOL [40, 44] dataset, which encompasses a diverse range of real-world low-light scenarios. LOL offers a rich collection of images captured under challenging lighting conditions, facilitating the training and evaluation of models for low-light image enhancement tasks. The SID [4] and SMID [3] datasets are captured by camera with both short and long exposure to obtain low- and normal-light images separately. The SDSD [34] is constructed by a camera with an ND filter in both indoor and outdoor. Different from previous dataset, Adobe FiveK [1] use the image adjusted by several photographer experts as the ground truth rather than the noraml light image in the same scenario. All of the above mentioned datasets primarily focus on how to reconstruct normal-light images from a given low-light RGB image. The LED dataset we propose in this paper first extend LIE task to multimodal by introducing depth information, and LED can thereby advancing research about the multimodal low-light image enhancement task.

### 2.2 LIE Methods

In recent years, LIE has become a prominent research area in computer vision and image processing. Numerous methods have been proposed to address the challenges posed by low light conditions and improve the visibility and quality of images captured in such environments. Traditional approaches [8, 15, 27] often relied on histogram equalization and contrast stretching techniques, but these methods tend to introduce artifacts and may not effectively handle complex low light scenarios. Advanced techniques have emerged, leveraging deep learning (DL) and neural networks for low light image enhancement. Researchers have explored the use of convolutional neural networks (CNNs) [20] to learn complex mappings

between low- and normal-light images, enabling the generation of visually pleasing results. Additionally, attention mechanisms [33, 46] and generative adversarial networks (GANs) [7, 30] have been employed to capture and enhance important details in low light scenes. Traditional Retinex [19] theory is also combine with DL methods and used in some methods [19, 40, 46]. However, previous methods are designed exclusively for only RGB images. Our LEDN is the first method designed for multimodal low-light image inputs with depth information.

## 2.3 RGBD Datasets

Currently, multimodal-based methods have been popular researched to improve model performance [22, 35, 37, 38, 42]. In the realm of computer vision and scene understanding [41, 45], RGBD datasets play a pivotal role in advancing the capabilities of various multimodal applications. Previous RGBD datasets mainly focus on 3D computer vision tasks. Over the years, researchers have contributed significantly to the development of diverse and comprehensive RGBD datasets, facilitating the algorithms for tasks such as 3D object recognition [28], 3D scene understanding [12], and 3D semantic segmentation [9]. One famous dataset is the NYU Depth V2 [31] dataset, which comprises indoor scenes captured by a Kinect sensor, providing RGB images along with corresponding depth maps. Another prominent dataset is the SUN RGB-D [32] dataset, offering a vast collection of indoor scenes with precise annotations for object instances and room layouts. These datasets have not only spurred advancements in computer vision research but also served as benchmark resources for evaluating the efficacy of algorithms across a spectrum of RGBD tasks. The depth map is less affected by changes in lighting and texture loss, and our LED dataset is the first that try to introduce the advantages of depth information to solve the problem in the LIE task.

## 3 Dataset

**Table 1: Comparison of LED with different LIE datasets.**

| Dataset | #Sample | Depth | Type | Source |
|---|---|---|---|---|
| LOLv1 [40] | 500 | ✗ | Image | real |
| LOLv2-real [44] | 789 | ✗ | Image | real |
| LOLv2-syn [44] | 1,000 | ✗ | Image | synthe |
| SID [4] | 2,697 | ✗ | Image | real |
| SMID [3] | 404 | ✗ | Video | real |
| SDSD [34] | 150 | ✗ | Video | real |
| LED (ours) | 1,365 | ✔ | Image | real |

## 3.1 Collection

To construct LED, we start from building the camera application with ARKit on iOS. ARKit is a framework developed by Apple for creating augmented reality (AR) experiences on iOS devices. It enables developers to integrate immersive AR content into their applications, blending digital elements with the real-world environment captured by the device's camera. In this paper, we primarily utilized

ARKit to access the LiDAR device alongside the rear camera of the iPhone, enabling the simultaneous capture of depth information during photography.

Unlike conventional binocular RGBD cameras or Time-of-Flight (ToF)-based cameras, iPhone's LiDAR employs pulsed laser beams, which is more stable and allowing for longer-range capture. According to ARKit's official documentation, the LiDAR on the iPhone can reach distances of approximately 5 meters. For capturing low-light images, similar to prior research [3, 4, 40], we adjust the exposure time and ISO of camera using iOS API to decrease brightness and simulate low-light conditions. For each scene, our data collection process involves: (1) fix the phone in place; (2) capture normal-light image with default camera configuration and record depth map at the same time; (3) adjust camera parameters and capture image under low-light conditions.

## 3.2 Quality Control

To ensure each image contains meaningful depth information, during capture, we ensured that the nearest object to the camera is at a maximum distance of 3 meters from the camera. And to enhance the balance of our dataset, we capture scenes that included both indoor and outdoor environments and controlling their quantities to be roughly equivalent. Additionally, to maximize data diversity, we capture only one image per scene, and we introduce randomness in adjusting exposure time and ISO, minimizing the chances of the model exploiting patterns. Moreover, our scene captures are also distributed across various time periods throughout the day. After collecting all data, because the maximum resolution of the depth map on the iPhone is only supported up to $768 \times 576$, we uniformly resize the resolution of both low-light and normal-light images to match the size of the depth map. We also perform Min-Max normalization on the depth map. Table 1 shows the comparison between our LED and previous LIE datasets. It can be found that LED is currently the only one that supports multimodal low-light image enhancement with additional depth information.

## 4 Methodology

### 4.1 Overview

Figure 2 shows the overall pipeline of our proposed LEDN. LEDN is a encoder-decoder based neural network in which we first use encoder layers to encode and down-sample the input low-light image, and then use the same number of decoder layers to decode and up-sampling the hidden states layer by layer. We also add a skip connection between the encoder and decoder in the same layer inspired by U-Net [29]. Finally, an RGB prediction layer is applied to estimate the color under normal light conditions.

In LEDN, we introduce three additional components to incorporate depth information into low-light image restoration. First, before input the orignal low-light image into image encoder, we use a Depth Boundary-Aware (DBA) module, based on the Sobel algorithm [16], to extract edge information of objects from the depth map, obtaining a binary mask image, and then concatenate it with the original image to impart edge-awareness to the input information. In low-light images, the edges of objects are often unclear, but the depth map, despite lacking color and lighting information, provides accurate edge information. This edge information can be

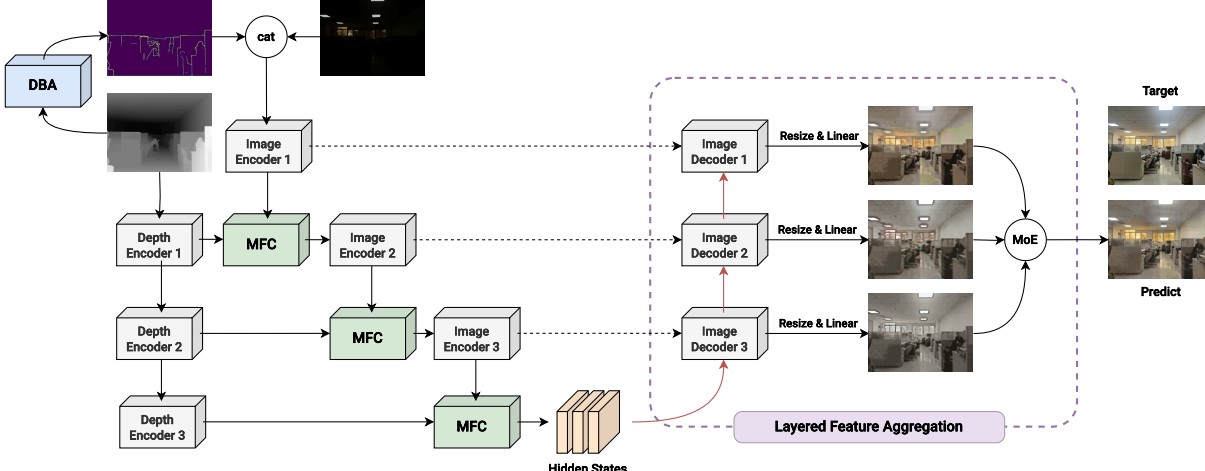

**Figure 2: Overall pipeline of LEDN. "DBA" means Depth Boundary-Aware module, "MFC" means Multimodal Fourier Convolution module, "MoE" means Mixture-of-Experts layer. The red line means up-sampling convolution.**

used to assist in the restoration of the image. The details of DBA are presented in Section 4.2.

Second, in addition to image encoders, here we employ the same number of depth encoders to encode depth map layer by layer. And we design a Multimodal Fourier Convolution (MFC) module to fuse image and depth features in the same layer, thus to integrate depth information into the RGB image. The details of the MFC module are described in Section 4.3.

The last major innovation of our model is the Layered Feature Aggregation (LFA) module. In LFA, instead of only using the output of the last decoder as the predicted restored image, we use the outputs of all three decoder layers to obtain the normal-light image. Since each layer from the bottom layer to the top layer has a different receptive field, and each receptive field is crucial for image restoration, here we employ a Mixture-of-Experts (MoE) to aggregate features from these three layers, resulting in the final output, which is detailed described in Section 4.4.

## 4.2 Depth Boundary-Aware Module

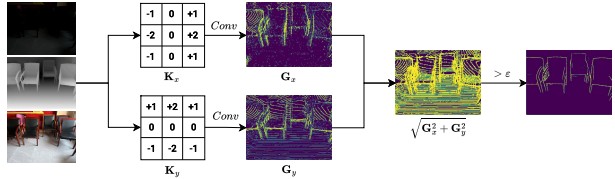

**Figure 3: Pipeline of Depth Boundary-Aware module.**

As described in Section 4.1, compared to low-light images with blurry boundaries, depth maps, unaffected by lighting conditions, can provide clear object edge boundaries. Thus, we can extract edges from the depth map to enhance the original image. To ensure computational efficiency while maintaining accuracy, we have employed the Sobel kernel [16] as the key component of our Depth

Boundary-Aware (DBA) module. The Sobel operator is a widely used image edge detection algorithm designed to identify intensity changes, particularly at edge locations within an image. The operator consists of a pair of $3 \times 3$ convolution kernels ($\mathbf{K}_x$ and $\mathbf{K}_y$) as shown in Figure 3. The kernels can be applied separately to the input image $\mathbf{I} \in \mathbb{R}^{3 \times H \times W}$, to produce separate measurements of the gradient component in each orientation. It operates through convolution, sliding a Sobel kernel across the image to detect gradients in both the horizontal and vertical directions to obtain the gradient map $\mathbf{G}_x$ and $\mathbf{G}_y$ as follows:

$$\mathbf{G}_x = \text{Conv}_{\mathbf{K}_x}(\mathbf{I}), \quad \mathbf{G}_y = \text{Conv}_{\mathbf{K}_y}(\mathbf{I}). \quad (1)$$

$\mathbf{G}_x$ and $\mathbf{G}_y$ can then be combined together to find the absolute magnitude of the gradient $\mathbf{G} \in \mathbb{R}^{1 \times H \times W}$ and also the orientation of that gradient at each pixel, which can be formulated as:

$$\mathbf{G} = \frac{\sqrt{\mathbf{G}_x^2 + \mathbf{G}_y^2}}{\text{Max}(\sqrt{\mathbf{G}_x^2 + \mathbf{G}_y^2})}. \quad (2)$$

where $\text{Max}()$ selects the biggest element value to conduct normalization. Since a high spatial frequency usually corresponds to the edge, we then use the gradients in both the horizontal and vertical directions to calculate the edge mask $\mathbf{E} \in \mathbb{R}^{1 \times H \times W}$ as follows:

$$\mathbf{E}_{i,j} = \begin{cases} 1, & \text{if } \mathbf{G}_{i,j} > \varepsilon \\ 0, & \text{otherwise} \end{cases} \quad (3)$$

where $\varepsilon$ is a threshold. After obtaining the binary edge mask, we concatenate it with the original image along the RGB dimension. Compared to parameterized methods based on deep learning, there is no parameter that need to be trained in DBA. This allows it to run at the fastest speed and meanwhile find high-quality edges, ensuring the effectiveness and efficiency.

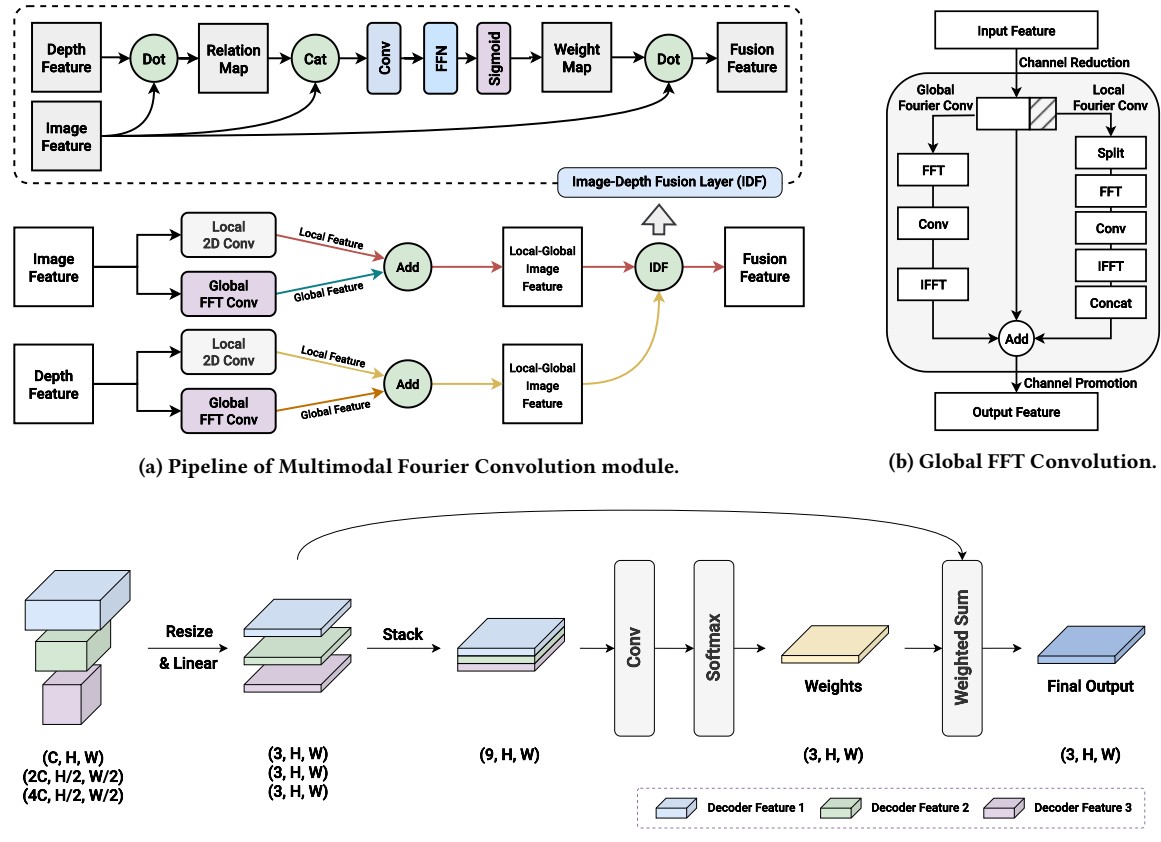

(a) Pipeline of Multimodal Fourier Convolution module.

(b) Global FFT Convolution.

(c) Pipeline of MoE layer.

**Figure 4: More details about LEDN.**

## 4.3 Multimodal Fourier Convolution Module

Due to the complexity and multi-scale nature of the image and depth data $\mathbf{D} \in \mathbb{R}^{1 \times H \times W}$, it is essential to effectively capture both local and global features before fuse them together. Therefore, we employ two different convolutional approaches to process the input images and depth features before combination. Figure 4a illustrates the structure of our Multimodal Fourier Convolution (MFC) module. Specifically, the first *Local 2D Convolution* (L2C) is a regular 2D convolution, utilized to extract local features. The second *Global FFT Convolution* (GFC) is a global convolution enhanced with Fourier Convolution (FC) based on Fast Fourier Transform (FFT), aimed at capturing global feature information. Since the convolution on frequencies impacts all pixels, we can thus achieve a global-level receptive field. Here, we implement GFC similar to the Spectral Transformer [6] as shown in Figure 4b.

In GFC, for input features, we first utilize a convolution layer to reduce their dimensionality to speed up the subsequent calculation. Then, for the processed features, the global FC branch on the left first performs FFT on the input features to convert them into the frequency domain. Then, it conducts convolution on the real and imaginary parts separately in the frequency domain. At last, the inverse FFT (IFFT) is used to return the features back to the spatial domain. On the right local FC branch, before performing FC, it

divides the features into several parts and applies FC separately to each of these parts, and then concatenates the resulting features at the end. This division ensures that convolutions occur only within each segmented feature block and achieves segment-level perception. Finally, we add the original features to the features obtained from the two branches to obtain the final output feature.

With L2C and GFC, we are able to obtain image and depth feature with both local and global information by adding the two kinds of features together, which are formulated as:

$$\mathbf{I}_h = \mathsf{L2C}(\mathbf{I}) + \mathsf{GFC}(\mathbf{I}), \quad \mathbf{D}_h = \mathsf{L2C}(\mathbf{D}) + \mathsf{GFC}(\mathbf{D}). \quad (4)$$

where $\mathbf{I}_h \in \mathbb{R}^{C \times H \times W}$ and $\mathbf{D}_h \in \mathbb{R}^{C \times H \times W}$ are hidden states with $C$ channels for $\mathbf{I}$ and $\mathbf{D}$, separately. Those two features are then input into the Image-Depth Fusion (IDF) layer (Figure 4a). IDF first takes the image and depth feature as input and calculates the relation map by element-wise dot production. The image feature and the attention feature are then concatenated and followed by a channel attention block to obtain the weight map. The channel attention block consists of a convulsion layer, a feed-forward network layer, and a sigmoid layer to constrain the weight to 0 to 1. Finally, the weight map is used to enhance the image feature by dot production with the initial image feature. The overall process can be formulated

as follows:

$$\mathbf{F} = \texttt{Sigmoid}(\texttt{FFN}(\texttt{Conv}([\mathbf{I}_h \cdot \mathbf{D}_h; \mathbf{I}_h]))) \cdot \mathbf{I}_h. \quad (5)$$

where $\mathbf{F} \in \mathbb{R}^{C \times H \times W}$ is the fusion feature that combined RGB and depth features through the above operations.

## 4.4 Layered Feature Aggregation Module

In the Layered Feature Aggregation (LFA) module, we try to combine the outputs from all three decoder layers to better restore the image from hidden states. For the three output hidden features of decoder side $\mathbf{O}_1, \mathbf{O}_2, \mathbf{O}_3$ that shape $[C, H, W]$, $[2C, H/2, W/2]$ and $[4C, H/4, W/4]$ respectively, from $\mathbf{O}_1$ to $\mathbf{O}_2$, $\mathbf{O}_2$ to $\mathbf{O}_3$, the receptive field area of features increases doubly each time. This leads to features that at different levels are able to focus on key features from different aspects. Since for each pixel, restoring its normal-light RGB value is not only related to itself but also to its surrounding pixels, incorporating features with various receptive field sizes enables each pixel to capture coarse-to-fine details, leading to a more precise restoration.

Furthermore, instead of simply reshaping and adding all the features together, here we employ a Mixture-of-Experts (MoE) layer to combine those features in a weighed manner. As shown in Figure 4c, we first resize the three hidden features in the shape $[H, W]$, then apply linear convolution to reduce their channels to 3, which corresponds to the R-G-B values, and then stack them together along the channel dimension to $\mathbf{O}_s \in \mathbb{R}^{9 \times H \times W}$ as follows:

$$\mathbf{O}_s = \texttt{Stack}([\mathbf{R}_1; \mathbf{R}_2; \mathbf{R}_3]), \ where \ \mathbf{R}_i = \texttt{Conv}(\texttt{Resize}(\mathbf{O}_i)). \quad (6)$$

For $\mathbf{O}_s$, we reduce its channel dimension to 3 without altering the width and height of the features by convolution. Then a softmax is applied along the channel dimension to obtain the weights $\mathbf{O}_w \in \mathbb{R}^{3 \times H \times W}$:

$$\mathbf{O}_w = \texttt{Softmax}(\texttt{Conv}(\mathbf{O}_s)). \quad (7)$$

For each element in $\mathbf{O}_w$, it contains three weights that will be applied from $\mathbf{O}_1$ to $\mathbf{O}_3$. Finally, we use $\mathbf{O}_w$ and $\mathbf{O}_s$ perform the weighted sum to obtain the final output $\mathbf{O} \in \mathbb{R}^{3 \times H \times W}$ which are formulated as:

$$\mathbf{O} = \sum_{i=1}^{3} \mathbf{O}_w^i \cdot \mathbf{R}_i. \quad (8)$$

where $\mathbf{O}_w^i \in \mathbb{R}^{1 \times H \times W}$ is per layer of $\mathbf{O}_w$ along the channel dimension.

## 4.5 Loss Function

We use Mean Squared Error (MSE) as the loss function and the goal is to minimize the MSE value between the predicted RGB value to the ground truth value, formulated as:

$$\text{MSE} = \frac{1}{n} \sum_{i=1}^{n} (y_i - \hat{y}_i)^2. \quad (9)$$

where $n$ denotes the number of samples, $y_i$ represents the actual values, and $\hat{y}_i$ corresponds to the predicted values.

## 5 Experiment

### 5.1 Experiment Settings

We implement LEDN using Pytorch [26] and train it with a 24GB RTX4090 GPU card. Throughout the training process, we resize the input images to a size of $400 \times 400$. For optimization, we employ Adam [17] with a momentum value of 0.9 and a weight decay of 1e-4. The base learning rate is set to 2e-4, while the batch size and the number of training epochs are set to 8 and 100, respectively. During training, we utilize the cosine learning rate decay. All the compared methods we reproduce are implemented using the original code released by the respective authors. We do not employ any data augmentation techniques or post-processing methods to refine all predictions for fair comparison. The LED dataset is randomly divided into training and validation sets in an 8:2 ratio, with 1,083 and 282 samples, respectively.

For evaluation, we use five types of metrics to assess model performance: Mean Squared Error (MSE), Peak Signal-to-Noise Ratio (PSNR) [11], Structural Similarity Index (SSIM) [36], Multi-Scale Structural Similarity Index (MS-SSIM) [39] and Learned Perceptual Image Patch Similarity (LPIPS) [48]. We use a pretrained AlexNet [18] as the feature extraction backbone [2] in LPIPS.

### 5.2 Quantitative Anaysis

We first conduct an experiment that compares our LEDN with previous SOTA methods on our LED dataset. Here, we choose methods over the past four years, including MIRNet [47], DeepLPF [24], SNR-Net [43], Restormer [46], EFINet [23], ChebyLighter [25], LEDNet [51], DNF [14], Retinexformer [2]. All experimental results are shown in Table 2. Our method achieves SOTA performance on all five metrics on the LED. Even compared to recent best method Retinexformer, our method is able to outperform it by a significant margin especially in PSNR (20.90 vs. 20.46), SSIM (0.481 vs. 0.469) and LPIPS (0.465 vs. 0.492). This proves that by using different modules in our method to combine the depth information, we can better restore the low-light images to normal light.

To further reveal the necessity of incorporating depth information into the LIE task, we try to enhance the previous SOTA methods by integrating depth information fusion into their models. The goal is to observe whether this integration could further improve accuracy compared to using only RGB images. We test several methods from Table 2 and enhance them with our MFC module to incorporate depth information into them. The results are shown in Table 3. It can be observed that in the vast majority of cases, MFC brings substantial performance improvements to different methods. This not only demonstrates the importance of introducing depth information for low-light image enhancement, but also validates the effectiveness of the MFC module we designed.

### 5.3 Ablation Study

We validate the impact of each module on LEDN by disabling one or more components and comparing the performance on the LED test set, and the results are shown in Table 4. To better demonstrate the effect of our proposed modules, we first concatenate the depth map onto the RGB image as RGBD input for comparison. The result

---

[2]https://github.com/richzhang/PerceptualSimilarity

**Table 2: Comparison of LEDN with different SOTA methods from LIE task on LED dataset. The top and second best results are highlighted in red and blue.**

| Method | Venue&Year | MSE↓ | PSNR↑ | SSIM↑ | MS-SSIM↑ | LPIPS↓ |
|---|---|---|---|---|---|---|
| EFINet [23] | TCSVT-2022 | 0.0198 | 17.82 | 0.443 | 0.734 | 0.607 |
| ChebyLighter [25] | ACMMM-2022 | 0.0201 | 17.80 | 0.445 | 0.729 | 0.485 |
| DeepLPF [24] | CVPR-2020 | 0.0211 | 18.06 | 0.414 | 0.749 | 0.698 |
| DNF [14] | CVPR-2023 | 0.0153 | 19.60 | 0.432 | 0.784 | 0.722 |
| MIRNet [47] | ECCV-2020 | 0.0125 | 19.80 | 0.414 | 0.779 | 0.680 |
| LEDNet [51] | ECCV-2022 | 0.0144 | 20.02 | 0.439 | 0.796 | 0.700 |
| SNR-Net [43] | CVPR-2022 | 0.0116 | 20.21 | 0.463 | 0.803 | 0.544 |
| Restormer [46] | CVPR-2022 | 0.0111 | 20.40 | 0.431 | 0.797 | 0.627 |
| Retinexformer [2] | ICCV-2023 | 0.0102 | 20.46 | 0.469 | 0.810 | 0.492 |
| LEDN (ours) | Review-2024 | **0.0100** | **20.90** | **0.481** | **0.822** | **0.465** |

**Table 3: Experimental results of previous SOTA methods enhanced with our MFC module. Performance improvements are shown in parentheses, where green numbers represent increased performance and red represent degraded performance.**

| Method | MSE↓ | PSNR↑ | SSIM↑ | MS-SSIM↑ | LPIPS↓ |
|---|---|---|---|---|---|
| EFINet [23] (+ MFC) | 0.0195 (0.0003) | 17.95 (0.13) | 0.446 (0.003) | 0.754 (0.020) | 0.611 (0.004) |
| DeepLPF [24] (+ MFC) | 0.0130 (0.0081) | 19.74 (1.68) | 0.457 (0.043) | 0.793 (0.044) | 0.565 (0.133) |
| DNF [14] (+ MFC) | 0.0121 (0.0032) | 20.02 (0.42) | 0.455 (0.023) | 0.802 (0.018) | 0.625 (0.097) |
| MIRNet [47] (+ MFC) | 0.0137 (0.0012) | 19.47 (0.33) | 0.484 (0.070) | 0.806 (0.027) | 0.507 (0.173) |
| LEDNet [51] (+ MFC) | 0.0107 (0.0037) | 20.60 (0.58) | 0.458 (0.019) | 0.820 (0.024) | 0.632 (0.068) |
| SNR-Net [43] (+ MFC) | 0.0111 (0.0005) | 20.34 (0.13) | 0.467 (0.004) | 0.801 (0.002) | 0.598 (0.046) |

**Table 4: Ablation studies on different modules in LEDN. The first two rows respectively show the results of using RGB and RGBD image as input without using three modules.**

| DBA | MFC | LFA | MSE↓ | PSNR↑ | SSIM↑ | MS-SSIM↑ | LPIPS↓ |
|---|---|---|---|---|---|---|---|
| ✗ | ✗ | ✗ | 0.0147 | 18.97 | 0.448 | 0.779 | 0.495 |
| ✗ | ✗ | ✗ | 0.0146 | 18.89 | 0.460 | 0.788 | 0.500 |
| ✔ | ✗ | ✗ | 0.0137 | 19.24 | 0.463 | 0.796 | 0.494 |
| ✗ | ✔ | ✗ | 0.0104 | 20.75 | 0.473 | 0.811 | 0.492 |
| ✗ | ✗ | ✔ | 0.0135 | 19.31 | 0.462 | 0.789 | 0.473 |
| ✔ | ✔ | ✗ | 0.0104 | 20.76 | 0.475 | 0.817 | 0.485 |
| ✔ | ✗ | ✔ | 0.0127 | 19.42 | 0.457 | 0.784 | 0.477 |
| ✗ | ✔ | ✔ | 0.0101 | 20.81 | 0.474 | 0.816 | 0.469 |
| ✔ | ✔ | ✔ | **0.0100** | **20.90** | **0.481** | **0.822** | **0.465** |

is shown in the second row of the Table 4. It can be observed that directly using RGBD image did not significantly improve the model's performance. We think this may because the RGB values of low-light images are relatively small, typically much less than 0.1, while the normalized depth values are uniformly distributed between 0 and 1. This results in a huge disparity between these two kinds of value, making it difficult for the model to utilize depth information effectively to reconstruct the image.

Then we compare the model performance with and without our three modules. First, it can be observed that by extracting edge information and concatenating it onto the RGB image, we are able to enhance the model's performance, which reveals not only the importance of edge information in LIE task but also the effectiveness of DBA. Furthermore, when using only one module, MFC has the most significant impact among the three modules. Compared to the basic network without any specific module, it increases the PSNR accuracy by 1.78. This demonstrates the effectiveness of the feature fusion approach employed by MFC, and meanwhile proving that introducing depth information into the LIE task is useful. LFA is the second most useful module, while DBA, although not as significant as the other two, still contributes to improving the model accuracy to some extent. Another interesting finding is that when using LFA, we are able to get a better LPIPS score compared to other two modules, which proves that the combination of different decoders' outputs can help to generate normal-light image that is preferred by humans. The model that simultaneously utilizes all three modules achieves the highest accuracy, which indicates that each module in LEDN makes a positive contribution to overall accuracy.

## 5.4 Qualitative Analysis

*5.4.1 Comparison.* To more intuitively compare the performance differences between our approach and the previous methods, we select several samples from the test set and visualize the reconstruction results of each model in Figure 5. We highlight some significant differences with the red boxes. Through comparison,

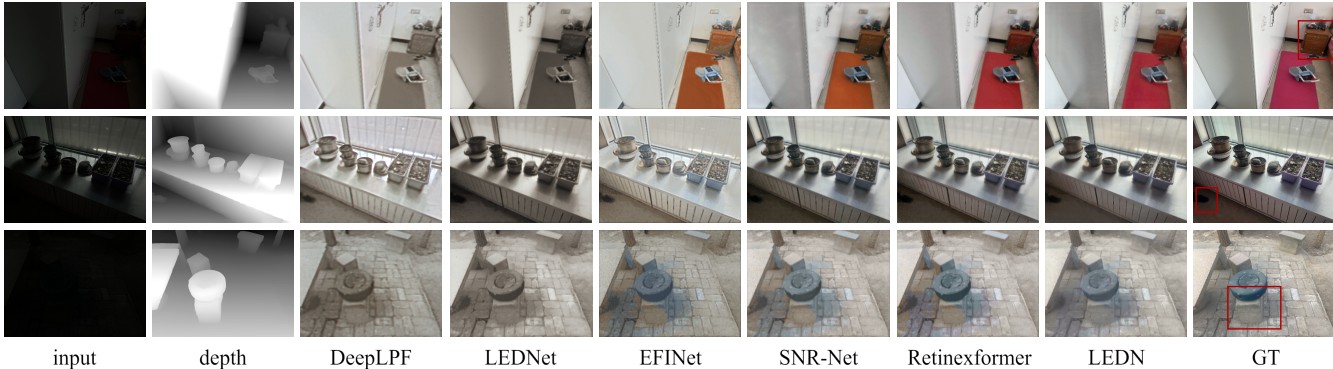

| input | depth | DeepLPF | LEDNet | EFINet | SNR-Net | Retinexformer | LEDN | GT |

**Figure 5: Results on LED dataset. Our method effectively improves image visibility while preserving correct color, especially in extremely dark areas.**

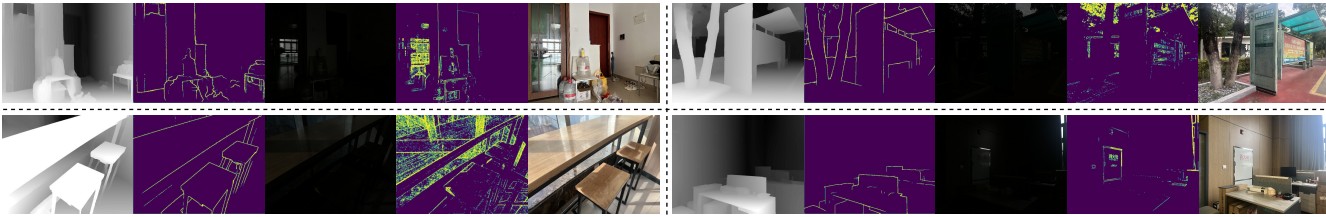

**Figure 6: Four samples of DBA module on LED dataset. From left to right are: (1) depth map; (2) boundary mask of depth map; (3) low-ligth image; (4) boudary mask of low-light image; (5) normal-light image.**

it can be observed that, compared to previous methods, our approach performs better on the edges of objects, and our model's reconstruction results are closest to the ground truth, especially in particularly dark areas. For instance, in the first sample, the shadow color produced by our method in the lower right corner of the bed cabinet is the lightest, whereas other methods, like Retinexformer, generate darker black shadows that do not exist in the actual image. In the second sample, for the slippers in the lower left corner, our model reconstructs the clearest outline by leveraging depth map information. Similarly, in the third sample, the boundaries between the stone stool and the stone floor in our generated images are more distinct, due to the boundary information extracted from the depth map. Moreover, the overall color of our reconstructed images is closer to the real scene. This demonstrates that by leveraging depth information, our method exhibits better robustness for low-light images under different lighting and environmental conditions.

*5.4.2   **Visualization of DBA**.* To visually demonstrate the role of the DBA module more intuitively, we utilize the DBA module to extract the corresponding boundary masks for depth maps and low-light images of four samples. These masks are displayed in Figure 6. As shown in this figure, in these four samples, the boundaries extracted from the depth maps are much clearer and contain less noise than those extracted directly from low-light images. For example, in the first and last samples, the boundaries in the low-light images are only clear in the bright areas on the left side, while in the dark areas on the right side, effective boundary information cannot be extracted. On the contrary, depth maps, unaffected by lighting

conditions, can extract most of the prominent object boundaries. Moreover, by observing the second and third samples, it can be noticed that the noise present in low-light images is reflected in the extracted boundary information, leading to dense and erroneous boundary information in some areas. For comparison, this confirms the effectiveness of our DBA module in extracting boundaries from depth maps and emphasizes the potential of improving the performance of low-light image enhancement models by focusing on the boundaries using additional depth information.

## 6   Conclusion

This paper presents a first-of-its-kind study of depth-based multimodal low-light image enhancement task. To facilitate further research in this area, we have created a first-of-its-kind comprehensive dataset called LED, which consists of 1,365 samples, each with a low-light image, a normal-light image, and a corresponding depth map. Additionally, we propose a novel network called LEDN, which incorporates feature fusion between RGB images and depth map to improve low-light image enhancement. Through extensive experiments and detailed analysis, we demonstrate that LEDN outperforms existing methods in LIE tasks with additional depth information on the LED dataset, which also proves the necessarily of leveraging depth map in LIE. Moreover, by combining our designed module with the previous method, we prove the effectiveness and flexibility of our approach, showing its ability to easily integrate with other models. We hope that our dataset and method can contribute to the advancement of this field in the future.

## Acknowledgments

We would like to thank all the reviewers for their valuable suggestions, which helped us improve the quality of our manuscript. And Dongyuan Li acknowledges the support from the China Scholarship Council (CSC).

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
