# OpenReview forum: "Multimodal Low-light Image Enhancement with Depth Information"
_acmmm.org/ACMMM/2024/Conference — MM2024 Poster_

### Official Review · Reviewer_GGwH · 2024-04-27

**Rating:** 6
**Confidence:** 3

**Summary:**

The paper presents a novel task of multimodal low-light image enhancement (MLIE) by leveraging depth information captured from iPhone LiDAR sensors, along with RGB images. The authors construct a new large-scale dataset called LED, comprising 1,365 samples of low-light RGB images, normal-light RGB images, and corresponding depth maps. Additionally, they propose a method named LEDN that incorporates depth information into a deep learning architecture for enhancing low-light images. Extensive experiments and ablation studies demonstrate the effectiveness of LEDN compared to previous state-of-the-art methods on the LIE task.

**Strengths:**

- Novelty: This work introduces a new problem formulation and dataset for multimodal low-light image enhancement, which is a practical and innovative direction in the field of image restoration.

- Dataset Creation: The authors have put significant effort into collecting the LED dataset using iPhone LiDAR sensors, ensuring diversity in scenes and lighting conditions. The dataset is a valuable contribution to the research community.

- Technical Approach: The proposed LEDN method is well-designed, incorporating depth information through three novel modules (DBA, MFC, and LFA) in a principled manner. The authors have carefully considered how to effectively integrate depth cues into the low-light image enhancement pipeline.

- Evaluation: The experimental results and ablation studies are comprehensive, demonstrating the effectiveness of the proposed approach compared to existing state-of-the-art methods on the LIE task. The qualitative and quantitative analyses are convincing.

- Clarity: The paper is well-written, well-organized, and easy to follow. The authors have clearly explained the motivation, technical details, and experimental findings.

**Limitations:**

While the paper presents a novel and well-executed work, there are a few potential suggestion:

- Generalization: The dataset and method are specifically tailored to iPhone LiDAR sensors. It would be beneficial to explore the generalization capabilities of the proposed approach to other depth sensing modalities or imaging devices.

- Computational Complexity: The paper does not provide an analysis of the computational complexity or runtime performance of the proposed LEDN method, which could be crucial for real-time applications or resource-constrained devices.

- Depth Map Quality: The quality and accuracy of the depth maps captured by the iPhone LiDAR sensor might vary depending on the scene complexity, lighting conditions, or distance from the camera. It would be interesting to investigate the robustness of the proposed method to depth map noise or inaccuracies.

**Suitability:**

3

---

### Official Review · Reviewer_j71b · 2024-05-16

**Rating:** 5
**Confidence:** 3

**Summary:**

This paper introduces a novel approach to enhancing low-light images by integrating depth information, leveraging advancements in mobile devices. It presents a comprehensive dataset named LED (Low-light Image Enhanced with Depth Map), consisting of 1,365 samples that include low-light images, normal-light images, and depth maps. By focusing on the potential of depth information for image restoration, the research identifies how this additional modality can significantly improve the accuracy of image enhancement. Detailed experiments and ablation studies demonstrate the efficiency of the proposed multimodal method, providing insights for advancing low-light image enhancement techniques.

**Strengths:**

1. The use of depth maps as a supplementary modality to provide contour features for enhancing low-light image restoration is an innovative and practical approach.
2. Achieve SOTA performance.
3. Extensive experimental evaluation is provided.
4. Open-source the both code and dataset.

**Limitations:**

1. The application scope is limited by the LiDAR's effective range of 3 meters, which might be a constraint imposed by the current LiDAR technology designed primarily for indoor use on mobile devices.

2. The method's reliance on LiDAR for depth information may limit its applicability, as many mobile devices utilize stereo cameras to obtain depth maps, which might not be compatible with the proposed approach.

**Suitability:**

3

---

### Official Review · Reviewer_YcaK · 2024-05-20

**Rating:** 1
**Confidence:** 3

**Summary:**

The article proposes a method for low-light image enhancement using depth images as texture information reference. The completion of the article is reasonable and the decomposition experiments are sufficient. And a low-light dataset containing depth information is proposed.

But what I am more concerned about is that the proposed method is limited in actual use, and lacks comparison with commonly used low-light data sets, which lacks fairness.

**Strengths:**

1 The article uses depth information to complete the low-light enhancement task, which is an interesting point of view.

2 The Multimodal Fourier Convolution Module proposed in the article is an innovative component.

**Limitations:**

What I am more concerned about is that the proposed method is highly restrictive in practical use.

1 The article mentioned that when the data set is collected, depth information is obtained by adjusting the intensity of light.

“we adjust the exposure time and ISO of camera using iOS API to decrease brightness and simulate low-light conditions.”

So when it is actually used, how to obtain effective depth information is a contradictory problem. It is obvious that the article did not deal with this problem, but only conducted various experiments on the datasets produced by itself.

2 The current common low-light enhancement datasets lacks high-quality depth information, and the current article is difficult to test and compare with SOTA methods.

My small suggestion, a potential alternative is to use a trained deep network to predict the depth map of the low-light image based on the Gama transform for inference.

3 Figure 5 does not have enough advantages compared to other methods.

4 Figure 6. The texture information extracted from the depth image is smooth enough. However, this is obvious because the depth map itself has ignored most of the detailed information. But for the task of low-light enhancement, the essence is to effectively adjust pixels of different brightness, that is, to darken high-brightness pixels and brighten low-light pixels. But the texture of low-light image pixels is sparse and discontinuous.

Such as light strips in the image, reflections produced by the glass, etc. But the depth information is lost in the texture of these special pixels.

**Suitability:**

2

---

### Official Review · Reviewer_onra · 2024-05-22

**Rating:** 5
**Confidence:** 3

**Summary:**

Incorporating depth into image restoration is a promising research area. this paper introduce a multimodal low-light enhancement task using depth data and establish the" Low-light Image Enhanced with Depth Map" dataset with 1,365 samples of low-light, normal-light images, and depth maps. To leverage depth information，the paper also design a corresponding multimodal method, which can processes the input images and depth map information simultaneously to generate the predicted normal-light images. Experimental shows the efficiency of the method .

**Strengths:**

This paper is written  clearly and easy to follow, presenting novel work. The primary contribution lies in the establishment of a dataset named LED (Low-light Image Enhanced with Depth Map), along with the proposal of a benchmark algorithm for low-light image restoration utilizing depth information. Although most of the modules used in the algorithm model are based on existing technologies, the author has designed them flexibly. The effectiveness of the method is thoroughly demonstrated through experiments.

**Limitations:**

1.Image degradation encompasses various types. In addition to the field of low-light image enhancement, it is necessary to introduce datasets and methods that combine other types of degradation with modalities.
2.This article uses some novel terms to describe the algorithms, such as the Depth Boundary-Aware (DBA) module and Mixture-of-Experts (MoE), but in fact, these are traditional methods, and the MoE is not very similar to those used in large models. Is it necessary to name them in this way?
3.There is an error in some of the descriptions, such as "RBG" in line 370.
4.Does the use of iPhone for creating the dataset in this paper introduce any device bias? Additionally, while the depth range for data collection in this dataset is within 5 meters, some scenarios such as autonomous driving involve depths greater than 5 meters. Has this been considered in the algorithm or dataset?

**Suitability:**

3

---

### Meta-Review · Area_Chair_rjTM · 2024-07-03

**Recommendation:** Accept (Poster)
**Confidence:** 4

**Metareview:**

This paper receives one Accept, two weak accepts and one reject. The reviewer YcaK thought the depth information was obtained by changing the ISO. In the rebuttal, the authors provided clarified this. Though the proposed method has obvious limitation pointed out by the reviewers YcaK, j71b and GGwH, this could be further investigated in the future work. Based on the recommendation of all the reviewers, the meta review made the final decision.